# The Diverse Transformer (Trf) Protein Family in the Sea Urchin *Paracentrotus lividus* Acts through a Collaboration between Cellular and Humoral Immune Effector Arms

**DOI:** 10.3390/ijms22136639

**Published:** 2021-06-22

**Authors:** Iryna Yakovenko, Asaf Donnyo, Or Ioscovich, Benyamin Rosental, Matan Oren

**Affiliations:** 1Department of Molecular Biology, Ariel University, Science Park, Ariel 40700, Israel; ireniren.y@gmail.com (I.Y.); donnyo9246@gmail.com (A.D.); or.ioscovich@gmail.com (O.I.); 2The Shraga Segal Department of Microbiology, Immunology and Genetics, Regenerative Medicine and Stem Cell Research Center, Ben-Gurion University of the Negev, Beer Sheva 84105, Israel; rosentab@post.bgu.ac.il

**Keywords:** *Paracentrotus lividus*, Transformer, Trf, 185/333, invertebrate immunity, phagocytosis, model organism

## Abstract

Sea urchins are long-living marine invertebrates with a complex innate immune system, which includes expanded families of immune receptors. A central immune gene family in sea urchins encodes the Transformer (Trf) proteins. The Trf family has been studied mainly in the purple sea urchin *Strongylocentrotus purpuratus*. Here, we explore this protein family in the Mediterranean Sea urchin *Paracentrotus lividus*. The *PlTrf* genes and predicted proteins are highly diverse and show a typical Trf size range and structure. Coelomocytes and cell-free coelomic fluid from *P. lividus* contain different PlTrf protein repertoires with a shared subset, that bind specifically to *E. coli*. Using FACS, we identified five different *P. lividus* coelomocyte sub-populations with cell surface PlTrf protein expression. The relative abundance of the PlTrf-positive cells increases sharply following immune challenge with *E. coli*, but not following challenge with LPS or the sea urchin pathogen, *Vibrio penaeicida*. Phagocytosis of *E. coli* by *P. lividus* phagocytes is mediated through the cell-free coelomic fluid and is inhibited by blocking PlTrf activity with anti-SpTrf antibodies. Together, our results suggest a collaboration between cellular and humoral PlTrf-mediated effector arms in the *P. lividus* specific immune response to pathogens.

## 1. Introduction

Sea urchins are long-living marine invertebrates (estimated life-span of ~15 years on average for *P. lividus* [1]) that are constantly exposed in nature to pathogens in the marine environment through direct contact with marine substrates, seawater, and consumed food. For protection, they have evolved a very sophisticated and robust innate immune system with multiple effector arms and a wide genome representation of immune genes [2]. The echinoid immune gene repertoire includes several expanded immune gene families that encode receptors and hemolymph proteins that participate in either cellular responses (e.g., Toll-like receptors, NOD-like receptors, scavenger receptor) or humoral responses (e.g., C-type lectins, complement components such as C3 and factor B) [3,4,5]. Despite the absence of vertebrate-like adaptive immunity, the large numbers of the germline-encoded immune proteins in sea urchins suggest that they may specifically recognize and act against a variety of pathogen-associated molecular patterns (PAMPs). One of the central immune gene families in sea urchin’s immune response encodes the Transformer (Trf) proteins.

The Trf protein family (formerly termed 185/333) was first identified in the California purple sea urchin *Strongylocentrotus purpuratus* [6]. Since then, more than 30 studies about the family in this species were published, and only one study in a different sea urchin species—*Heliocidaris erythrogramma* [7]. To date, no *Trf* genes or transcript sequences have been identified outside of the echinoid lineage, suggesting that Trf-based immunity is unique to sea urchins. In *S. purpuratus*, a robust Trf-mediated response was documented upon challenge with bacteria including *Escherichia coli* and *Vibrio diazotrophicus* [8,9] and with PAMPs including lipopolysaccharide (LPS), β-1,3-glucan and double-stranded RNA [10]. The response was characterized by a significant increase in *SpTrf* transcript and protein quantities [8,10,11,12,13] and in increasing Trf-expressing cell ratios of the whole coelomocytes population [14,15,16]. The SpTrf proteins are capable of binding to subsets of different antigens [17] and can augment phagocytosis and retard bacterial growth [18]. A recombinant SpTrf protein undergoes a shape transformation upon binding to antigens [16]. Consequently, the family name was changed from 185/333 to Transformer.

The *SpTrf* genes are 1.2–2 kb long and consist of two exons; a short leader and a second exon which encodes the functional protein [11]. The structure of the second exon is composed of a mosaic of conserved blocks of sequences, termed elements, that are either present or absent in different members of the family [11,12,19]. The defined combinations of the elements in a second exon are the primary source of diversity in gene sequence and size. However, additional diversification processes occur downstream, including post-transcription [20] and post-translation [6], resulting in a diversity of up to 260 different SpTrf proteins per individual as deduced from 2DE Western blot analysis [8,13]. Fifteen *SpTrf* genes were identified within two loci in the sequenced sea urchin [21]. However, the *SpTrf* loci are predicted to be highly unstable due to gene clustering and tandem sequence repeats [22,23] and the *SpTrf* gene repertoire shows high variability in the sea urchin population [21]. Furthermore, *SpTrf* genes may be subjected to regulated genomic diversification processes leading to gene deletions, duplications, and single point mutations (SNPs) in single cells [15] and that each cell may express a unique single *SpTrf* gene [9].

The Trf proteins are intrinsically unstructured in their native form and are predicted to adopt an α helical structure upon binding to a target [24]. Their predicted molecular weight based on the size of the messages ranged from 4 to 55 kDa in *S. purpuratus* [10] and from 8 to 39 kDa in *Heliocidaris erythrogramma* [7], while the actual protein range as observed on western blots is 30 kDa to >200 kDa in both sea urchin species [7,13,16]. This difference in size is attributed to the multimerization of the Trf proteins [16]. The primary structure of the Trf proteins consists of a glycine-rich region, a histidine-rich region and a C-terminal region [11]. The Gly-rich region contains the protein multimerization motif in its C-terminus, which mediates the Trf protein multimerization [17]. Recombinant peptides of the Gly-rich and His-rich Trf regions have both different and overlapping antigen-binding specificities [17]. The Trf proteins are expressed in a subset of the phagocytic class of sea urchin coelomocytes in at least two different expression patterns. In small phagocytes, Trf proteins are localized to the cell surface and in cytoplasmatic vesicles, whereas in large polygonal and discoidal phagocytes, the proteins were localized to perinuclear vesicles [7,16]. On the other hand, no Trf proteins were identified in the other cell types, including red and colorless spherule cells and vibratile cells [25]. Immunolocalization of SpTrf proteins in *S. purpuratus* histological sections identified cellular SpTrf protein expression in all major sea urchin organs, including axial organ, gut, esophagus, gonad, and pharynx [14], which is attributed to infiltrated coelomocytes [25].

Based on multiple studies in *S. purpuratus*, a rough model for Trf-mediated immune response was proposed [18,25]. According to the model, the Trf protein isoforms that are stored within the perinuclear vesicles of phagocytes are secreted to the coelomic fluid following the detection of bacterial challenge. The secreted Trf protein isoforms multimerize with the same/other isoforms upon binding to bacteria and may adapt an α helical shape. The proteins are also predicted to be involved in bacterial growth retardation [18]. It is not known how Trf-bound bacteria are recognized by the phagocytes; however, it was suggested that it may be facilitated through either an unknown receptor, multimerization with surface Trf proteins or through phosphatidic acid (PA) located in the plasma membrane. Trf protein binding to PA may cause membrane deformation and therefore may participate in phagocytosis [26].

To date, the *Trf* gene/protein family was characterized into only two sea urchin species—*S. purpuratus* and *H. erythrogramma.* Here, we characterize the *Trf* genes and proteins and the Trf-mediated immune response in a third sea urchin species—the Mediterranean Sea urchin *Paracentrotus lividus*. *P. lividus* is abundant in shallow subtidal areas and in tidal pools on rocks or seagrass meadows in the Mediterranean Sea and in the North-East Atlantic Ocean [27]. It is widely used in the food industry (e.g., [28]) as well as a model in scientific research, including the fields of developmental biology (e.g., [29]) and comparative immunology (e.g., [30]).

We report here that, similar to *S. purpuratus* and *H. erythrogramma*, *P. lividus* utilizes a diverse PlTrf protein repertoire as part of its innate immune response to pathogens. The *PlTrf* genes and the translated protein sequences, although phylogenetically distinct from *S. purpuratus* and *H. erythrogramma*, bear the same basic structure as *SpTrf* and *HeTrf* genes and proteins. In this study, we used three anti-SpTrf antibodies previously raised against the peptides within the N terminus, the central part and the C terminus of the mature Trf proteins [16]. Using three different approaches, we verified the cross-reactivity of these antibodies with PlTrf proteins. Western blot analyses show high PlTrf protein diversity among different *P. lividus* individuals. Furthermore, each of the sea urchins tested expressed a different Trf protein profile in the coelomocytes vs. the cell-free coelomic fluid (CF). Using fluorescence-activated cell sorting (FACS), we identify five different coelomocyte sub-populations with detectable membranal PlTrf expression. We also show that a challenge with heat-killed *E. coli* is followed by a sharp increase in the ratios of Trf-positive coelomocytes and that the *P. lividus* response to *E. coli* challenge is likely mediated by a specific set of PlTrf proteins. Lastly, we demonstrate that *P. lividus* cell-free CF is important for phagocytosis of *E. coli*, a process that can be inhibited by blocking membranal PlTrf function with the anti-Trf antibodies.

## 2. Results

### 2.1. The PlTrf Family in the Mediterranean Sea Urchin Paracentrotus lividus

For the initial characterization of the Trf family in *P. lividus,* we obtained a total of 94 *PlTrf* transcript and DNA sequences from the Transcriptome Shotgun Assembly (TSA) databases HACU01, GCZS01, GEDS01 and GFRN01 (NCBI) and Octopus database. Available online on request: http://octopus.obs-vlfr.fr/blast/oursin/blast_oursin.php (accessed on 15 March 2021), of which 70 sequences were unique after untranslated regions UTR sequences were removed (Appendix A). To understand the phylogenetic position of the *PlTrf* sequences, 25 *PlTrf* CDS sequences of different sizes were chosen (Appendix A) and aligned to an equal number of randomly chosen full *SpTrf* and *HeTrf* CDS sequences (Figure 1A). The alignment was used to generate a phylogenetic tree that showed separate clustering of the Trf sequences according to species with similar evolutionary distances among the clusters (Figure 1A). In contrast to the other two species, of which sequences showed homogeneous distribution, the *P. lividus* Trf cluster was sub-divided into three clades. This subdivision of *PlTrf* sequences may reflect segregation in function or may be caused by the limited sequence repertoire of the currently available sequences in databases from which the sequences were obtained. To date, no *Trf* sequences have been identified in non-echinoid echinoderm classes such as sea stars (class: Asterozoa) and sea cucumbers (class: Holothuroidea) nor in basal sea urchins such as pencil sea urchin *Eucidaris tribuloides* of the cidaroid order (Figure 1A), suggesting the Trf family is unique to higher sea urchins [25]. Within the echinoid lineage, in addition to Trf sequences from *S. purpuratus*, *H. erythrogramma*, and *P. lividus, Trf* sequences were also identified in *Strongylocentrotus franciscanus* and *Alocentrotus fragilus* [31]. A single *Trf*-like sequence from *S. intermedius* (genebank accession: ADO13494.1) was also reported.

To characterize the Trf primary protein sequences from *P. lividus*, we selected twelve full-length translated *PlTrf* CDS with unique sequence and element content. The translated PlTrf coding sequences were 220 to 308 amino acids (aa) in length, which was similar to the size range of the HeTrf protein sequences (257–349 aa) and the SpTrf sequences (276–354 aa). All PlTrf protein sequences had the typical signatures of Trf sequences, including the leader sequence, glycine-rich (Gly-rich) and histidine-rich (His-rich) regions and conserved sequence elements. However, the RGD motif (Arg-Gly-Asp cell-binding peptide motif), previously characterized in *S. purpuratus* [11], was absent from both PlTrf and HeTrf protein sequences suggesting that it is not an essential motif in PlTrf proteins. The translated protein sequences had a variable number of glycine residues in the Gly-rich region. The His-rich regions were also variable in the number of histidines, in which shorter versions contained nine histidines (element 20 in sequences PL001, PL005, PL006, PL009, PL023) and longer versions had up to 19 histidines (elements 19–20 in sequences PL10, PL011, PL015, PL018, PL021 and PL022) (Figure 1B). An alignment of all unique translated *PlTrf* CDS resulted in 22 elements that were 3 to 64 aa long and were identified based on gaps inserted into the alignment. The PlTrf elements contained between 3 and 64 aa. The PlTrf hydrophobic leader sequences contained 21 aa, and the most common leader sequence was MELKLILIIAIVAAITISA with variations at positions 5, 7, 9 and 16 (Figure 1B).

### 2.2. Anti-SpTrf Polyclonal Antibodies Cross React with PlTrf Proteins

Polyclonal SpTrf antibodies, provided by L. Courtney Smith (George Washington University), were originally raised against three synthetic peptides corresponding to relatively conserved sequences of the N-terminus, central part and C-terminus of the mature SpTrf proteins [16]. The three peptides for which the antibodies were raised are termed “66” (AHAQRDFNERRGKENDTER), “68” (GGRRGDGEEDTDAAQQIGDGLG), and “71” (GTEEGSPRRDGQRRPYGNR) based on rabbit numbers and are part of SpTrf sequence elements 1, 7 and 25 (see Figure 5B in Dheilly et al. 2009). Similar sequences (41–66% aa identity to the 66 region, 30–40% to the 68 region, and 70–88% to the 71 region) were identified in the *PlTrf* translated sequences within elements 1–3, 12–15 and 20–22 (Figure 2A). Since their development, the anti-SpTrf antibodies have been used extensively and have become an essential tool for studying the Trf protein family. To use these antibodies to study the PlTrf family in *P. lividus*, as was performed for the HeTrf family in *H. erythrograma* [7], we verified their cross-reactivity with PlTrf proteins based on three different approaches. First, we used a nickel (Ni)-column to enrich the PlTrf proteins from a whole coelomic fluid (CF) protein extract based on the binding of multiple histidine residues in the Trf proteins to Ni. The resulting WB showed different patterns of Ni-bound vs. unbound proteins (Figure 2B). Two major Trf protein sizes of ~120 and ~50 kDa for sea urchin #1 vs. ~250 and ~30 kDa in sea urchin #2 were identified in the Ni-bound fraction. In contrast, proteins mainly of ~12 and ~20 kDa in size were identified in the unbound fraction of Sea urchin #1 and were absent from Sea urchin #2, which had only >250 and several weak bands of smaller sizes in its unbound fraction (Figure 2B). These results are similar to previous results from Ni-column enrichment of SpTrf proteins that identified different SpTrf protein sizes in the Ni-bound and unbound fraction of the Ni-column assay [8,18]. The unbound PlTrf proteins may include family members with shorter his-rich regions, such as the proteins that would be encoded by sequences PL001, PL005, Pl006, PL009 (Figure 1B).

To test whether each of the three anti-SpTrf polyclonal antibodies (66, 68 and 71) had a similar binding pattern, we used the antibodies separately in WBs of coelomocytes, and CF protein extracts from the same individual sea urchin. Except for two differences in the coelomocyte fractions of the 68 and 71 anti-SpTrf blots, the resulted WB band patterns were identical for the three antibodies, suggesting that the anti-SpTrf antibodies show specific binding to the PlTrf proteins. The two differences were 1) a ~30 kDa band that was present in the coelomocyte fraction of the 68 anti-SpTrf blot, which replaced the ~20 kDa band that appeared in the coelomocyte fractions identified by the two other antibodies, and 2) an additional weak ~17 kDa band in the coelomocyte fraction of the 71 anti-SpTrf blot, which was not present in any of the other blots (Figure 2C). We speculated that these differences were due to the truncation and multimerization processes of specific membrane attached PlTrf proteins. It was also observed that the WB band patterns for the three anti-SpTrf antibodies were significantly different between the coelomocytes and the CF (Figure 2C). Because this difference between the cellular and humoral Trf fractions has not been characterized previously, we followed these results with additional analysis using additional sea urchins (see below).

In a third approach, to verify that the anti-SpTrf antibodies are bound to the PlTrf proteins, we implemented a targeted mass-spectrometry (MS) analysis on anti-SpTrf bound proteins from native *P. lividus* protein extracts and from the Ni-enriched protein fractions. Using the relevant WBs as a reference, SDS gel slices, located in the same region of the gel as the anti-SpTrf-positive high molecular weight bands, were sent for MS analysis. Results identified PlTrf peptide sequences from both native and Ni-enriched fractions. As expected, more PlTrf matches were obtained in the Ni-enriched fraction compared to the whole native extract (6 vs. 4). While matches of numerous conserved proteins from different sea urchins were identified, Trf protein matches were strictly of *P. lividus* (Table 1). The size range of all the PlTrf proteins identified by MS was 239–307 aa with a predicted molecular weight of 27.2 to 34.3 kDa and a predicted range of isoelectric points (pI) of pH 7.5 to 8.8. The presence of multiple Trf isoforms with many matches in the samples indicates a high confidence level for the presence of the PlTrf proteins in the samples from *P. lividus*.

### 2.3. P. lividus Coelomocytes and CF Have Different Subsets of Trf Proteins

Following our initial results showing different Trf protein profiles in coelomocytes vs. cell-free CF, we tested whether this pattern was consistent for other *P. lividus* individuals. WBs with the anti-SpTrf antibody mixture resulted in different Trf profiles of the coelomocytes and cell-free CF fractions of seven sea urchins both in Trf band content and relative band intensity (given that the general protein concentrations were the same for the coelomocyte and cell-free CF fraction samples for each animal). No consistent pattern was identified for the coelomocytes and cell-free CF fractions, except for a small band of ~20 kDa that was present in the cell-free CF fractions of all tested individuals but was absent or very weak in the coelomocyte samples (Figure 3). A strong ~35 kDa band was present in five of the seven individuals in either the coelomocytes (sea urchins 3, 4, 5) or cell-free CF (sea urchin 2, 7) fractions. Although the PlTrf proteins sampled for MS were of high molecular mass of 200–300 kDa (Appendix A), the largest PlTrf protein mass according to the MS analysis was ~34 kDa (Table 1). We, therefore, assume that all PlTrf bands of larger sizes represent multimerized proteins, similar to what was established for *S. purpuratus* and *H. erythrograma* ([7,8]).

### 2.4. Trf-Positive Cell Types in P. lividus

*P. lividus* coelomocytes are morphologically divided into phagocytes (filopodial and small), vibratile cells, red and colorless spherule cells [34], also termed amoebocytes [30]. To characterize the cell-surface PlTrf expression, we used FACS to sort Trf-positive coelomocytes based on their binding to the anti-SpTrf antibody mixture according to [15,34] (Appendix A). Using fluorescent microscopy, we identified cell surface Trf proteins in five immune coelomocyte sub-populations (Figure 4A). This is contrary to results from *S. purpurtus* that detected cell surface SpTrf proteins only in small phagocytes, whereas red spherule cells were specifically reported to lack SpTrf proteins [25]. Based on their distinct autofluorescence, red spherule cells could be analyzed separately as reported in [15,30,34,35], showing the presence of the Trf-positive red spherule cells (double-positive for APC-for autofluorescence and FITC-secondary antibody) compared to secondary antibody only control (APC-positive only, Figure 4B).

### 2.5. The Relative Abundance P. lividus Trf-Positive Coelomocytes Increase in Response to E. coli but Not to LPS or Sea Urchin Pathogen V. penaeicida Challenge

We previously showed that the percentages of *S. purpuratus* SpTrf-positive coelomocytes increase following the challenge with heat-killed *Vibrio diazitrophicus* bacteria [15]. In this study, we characterized the percent changes in the *P. lividus* Trf-positive coelomocyte following challenge with heat-killed *E. coli* bacteria, LPS, and heat-killed *P. lividus* pathogenic bacteria *V. penaeicida* [36] compared to percent changes in response to injection of sterile artificial CF (aCF). Although sea urchins were kept in isolated aquaria for more than 6 months before the experiment, their basal levels of Trf-positive coelomocyte percent changes were highly variable (13–65% of the total coelomocytes). In the first experiment, 10^6^ heat-killed *E. coli* bacteria or 1 μg of LPS per mL of CF were injected in 100 μL of sterile aCF into the coelom of the sea urchins of the two challenged groups (*n* = 3/group). An equivalent volume of pure aCF was injected into the control animals (*n* = 3). Coelomocytes (from 0.2–0.4 mL of CF) were collected for FACS analysis at day 0, just before the injection, and on days 1, 2, 13, and 43 post-injections. A significant (*p* < 0.05) percent change increase in the Trf-positive coelomocyte was observed in response to the *E. coli* challenge after day 1, which lasted through day 43 post-challenge (Figure 5A). On the contrary, the aCF-injected sea urchin controls and LPS-injected sea urchins showed a decrease in the percentages of Trf-positive coelomocyte on days 1 and 2 post-challenge, which was followed by an increase in percentages toward pre-injection levels in day 13 and 43 post-injection. The decrease in the Trf-positive coelomocyte percentages compared to the basal levels may have been due to insufficient time for coelomocytes to differentiate into Trf-expressing cells after removal of substantial 0.2–0.4 mL of sea urchin CF from their coelomic cavity. We therefore adopted a different experimental design, which included CF sampling starting from day 2 to allow enough time for Trf-expressing coelomocytes to accumulate. Indeed, when we followed this design, we did not observe a decrease in the Trf-expressing coelomocyte population in the unchallenged sea urchin controls. On the contrary, the injection of aCF in the control group resulted in a mild increase in the percentages of Trf-expressing coelomocytes (Figure 5B). This suggested that the decrease in Trf-positive cells in the first experiment was not related to the injections but rather to the timing of the sampling. We further speculate that the time needed for Trf-expressing cell progenitors to differentiate after depletion due to sampling was between one and two days. In the second experiment, we tested the Trf-mediated cellular response to challenge *V. penaeicida*, which may be a pathogen of *P. lividus* [36]. Heat-killed *V. penaeicida* cells (10^6^) per 1 mL of CF were injected into group 1 of sea urchins (*n* = 5), and equivalent volumes of aCF were injected into the negative control animals of group 2 (*n* = 5). No significant differences were observed between the challenged sea urchins and the control animals over 30 days of the experiment (Figure 5B). In both groups, a moderate percent increase in the Trf-expressing coelomocytes was observed on day 2 followed by a significant (*p* < 0.05) decrease to the basal levels on day 14 and another unexplained increase on day 30. However, a distinct, highly Trf-positive cell population appeared on day 2 in most of the experiment and control samples based on flow cytometry analysis (Appendix A), which was not present on days 14 and 30, suggesting Trf-based short-time response to both bacteria and aCF injections due to the tissue injury caused by the penetration of the needle.

### 2.6. A Subset of Trf Proteins Bind Live E. coli

Quite a few *S. purpuratus* studies have demonstrated SpTrf-mediated responses to different types of immune elicitors, including the expression and binding of subsets of SpTrf proteins [8,9,10,11,17,18,37]. Following the strong PlTrf cellular effector reaction to *E. coli*, we tested the profiles of the Trf proteins that bound to live *E. coli*. We used the cellular and cell-free CF fractions extracted from three sea urchins. WB analysis of the *E. coli*-bound proteins showed a conserved band pattern in both coelomocytes and cell-free CF fractions from the three sea urchins after 30 min of incubation with live *E. coli*. Most of the *E. coli* bound Trf proteins were confined to 5–9 bands of ~ 30 to ~60 kDa (Figure 6). The *E. coli*-bound protein band sizes were absent from the untreated protein extract taken from the same three sea urchins, suggesting they represent diverse Trf monomers or dimers that may be bound to *E. coli* molecules.

### 2.7. Phagocytosis in P. lividus Is Mediated through the CF and Is Partially Inhibited by Blocking Surface PlTrf Protein Activity

Two morphological types of phagocytes were described for *P. lividus*—filopodial and petaloid phagocytes [30,38]. The two phagocyte types are similar in size and contain long microfilament or cytoplasmatic projections of different morphology. We performed phagocytosis assays using coelomocytes that were incubated with fluorescent *E. coli* similar to [18]. *E. coli* were identified inside both *P. lividus* phagocyte types. The ratios of the phagocytes with internalized *E. coli* compared to the general coelomocyte population were 1 to 3%. Among these phagocytes, petaloid phagocytes were observed with a single internalized bacterial cell, whereas filopodial cells had multiple (up to 15) internalized bacteria (Figure 7A). The body fluid in many invertebrates has a major role in augmenting phagocytosis. To test if this is the case for *P. lividus* CF, we incubated *P. lividus* coelomocytes with either *E. coli* that were pre-incubated with CF (treatment A) or with untreated *E. coli* control (treatment B). Our results show almost two-fold higher phagocytosis rates by *P. lividus* phagocytes when *E. coli* were pre-incubated with CF compared to non-incubated bacteria (Figure 7B). It was reported that SpTrf proteins augment the phagocytosis of bacteria in *S. purpuratus* [18]. To test whether the membrane-bound Trf proteins participate in mediating phagocytosis in *P. lividus,* we used the anti-SpTrf antibody mixture to block the cell surface Trf activity before incubation with *E. coli* (treatment C). Indeed, bacteria-containing *P. lividus* phagocyted bacteria were found in significantly (*p* < 0.05) lower percentages in the coelomocytes that were pre-treated with anti-SpTrf antibodies (Figure 7B) compared with the equivalent untreated coelomocytes from the same sea urchins (Figure 7B, treatment A) (*n* = 3).

## 3. Discussion

Immune systems are one of the most diverse systems in multicellular organisms, both among species as well as within the population of a species and even among different cells of an individual. This is especially true for invertebrates that use many different ways to achieve protection against a great variety of pathogens in their surrounding environment (e.g., [39]). The Trfs are an example of unique proteins that participate in the effector arm of the innate immune response in sea urchins. To date, this family has been studied in *S. purpuratus* [25] as well as in *H. erythrogramma* [7]. Here, we characterize a family of Trf genes and proteins in the Mediterranean Sea urchin *P. lividus*, an echinoid model organism. Our results suggest that despite the differences among PlTrf, SpTrf, and HeTrf gene and protein sequences, the basic features and the mechanisms of action appear to be conserved. The PlTrf protein sequences vary in size and include typical Trf regions such as the characteristic hydrophobic leader, glycine-rich and histidine-rich regions, identifiable element patterns, and repeats that were characterized for the two other species. On the other hand, the RGD motif, which is located in the multimerization region, is absent from both *P. lividus* and *H. erythrogramma* and therefore may be unique to *S. purpuratus* and may not have a role in multimerization. To study the Trf-mediated response in *P. lividus*, we established a strong molecular tool, which is sometimes a challenging task in non-classical invertebrate models. We invested resources to validate the cross-reactivity of a potent anti-SpTrf antibody mix that was developed by the Smith group against conserved areas of the SpTrf proteins [16]. We used the validated antibody mix to characterize the PlTrf protein repertoire in cells and CF of different individual sea urchins to detect PlTrf-expressing coelomocytes, FACS analysis, and for the blocking the cell surface PlTrf activity in functional phagocytosis assays.

As in other sea urchin species, the PlTrf proteins vary among individuals, although they often share subsets of proteins of the same molecular weight. It was previously hypothesized that some SpTrf proteins multimerize through the glycine-rich multimerization region but that only monomers bind, lyse, or alter the shape of liposomes containing phosphatidic acid [26]. The PlTrf protein multimers vary greatly in size, suggesting several levels of multimerization. The Trf protein multimerization is irreversible and stable at high temperature in 2-mercaptoethanol, and SDS treatments, which is why multimers are observed on WBs. We identified two size categories of PlTrf protein monomers of approximately 20 and 34 kDa that are similar to the sizes of the SpTrf and HeTrf protein monomers. While the ~34 kDa monomer size was predicted based on the CDS sequences, the ~20 kDa monomer could not be predicted based on the available full-length primary sequences and may represent truncated proteins. In *S. purpuratus*, truncated Trf sequences that lack the histidine-rich region may serve as potential surveillance proteins [8]. This may also the case with the PlTrf as supported by the Ni-column isolation in which the ~20 kDa proteins were found in the unbound fraction from one of the sea urchins (Figure 2A).

Recent reports on the function of the SpTrf proteins suggest that they act through both cellular and humoral effector arms to augment phagocytosis [18,26]. Here, we identified significant differences between the PlTrf protein repertoires associated with the coelomocytes (cellular) compared to those in the cell-free CF (humoral) fraction of the coelomic fluid in each of the tested sea urchins. Differences are observed in the monomer and the multimer content of the two fractions. The smaller PlTrf monomers of ~20 kDa, which are consistently present in the cell-free CF, are either absent or almost absent from coelomocytes. On the other hand, the ~35 kDa monomers were present in either coelomocytes, cell-free CF, or both, depending on the sea urchin. Significant variability was observed in the PlTrf multimer content in all tested sea urchins. To better understand the functions of each of these Trf-mediated effector routes and the possible interplay between them, we studied each of the two fractions separately and tested how the absence or inhibition of the fractions affects the phagocytosis process.

In contrary to the cell surface localization of Trf proteins, which was restricted to small phagocytes (petaloid coelomocytes) in *S. purpuratus* [25], we identified at least five morphological subpopulations of coelomocytes with cell surface-bound PlTrf proteins. These differences between *S. purpuratus* and *P. lividus* may be due to differences in Trf activity between the species. However, they may also be due to the use of FACS to sort live coelomocytes, which enabled sensitive detection prior to imaging. An increase in membranal Trf-expressing coelomocyte percentages was recorded in *S. purpuratus* in response to LPS [16] and *V. diazotrophicus* [9,15]. We followed a similar experimental design to test first the effect of LPS and heat-killed *E. coli*. We were surprised to see that the results obtained were different from *S. purpurtus*. LPS does not influence the percentage of PlTrf-expressing coelomocytes, whereas there is a sharp increase in PlTrf-expressing cells in response to *E. coli*. This suggests that LPS might not be the main target of the PlTrf proteins but rather different molecules on the *E. coli* surface. We also used heat-killed *V. penaeicida*, a putative *P. lividus* pathogen, which is similar to *V. diazotrophicus*, that was used to challenge *S. purpurtus*. Unexpectedly, *V. penaeicida* does not have an effect on the PlTrf-expressing coelomocytes, suggesting that this species may escape the surveillance by the PlTrf proteins.

Specificity is a key feature in immune systems because it enables optimal recognition of pathogens within a variable environment and prevents self-destruction through autoimmunity [40]. Although this attribute is classically linked with adaptive immunity, a high degree of specificity may characterize certain types of innate immune systems and innate immune receptors. The great protein diversity of the Trf family within sea urchin populations [13] and the unique single-gene expression in individual coelomocytes [9] allow a high degree of specificity. Indeed, SpTrf specificity was demonstrated in the specific gene-expression response to different elicitors [8,9,10,18], the semi-specific binding of a recombinant SpTrf protein [17] and the selective PlTrf-mediated cellular response to different types of immune challenges, as shown here. By incubating live *E. coli* with coelomocytes or cell-free CF, we profiled the specific *E. coli* bound PlTrf protein profile. The profiles included the same specific sets of PlTrf proteins, regardless of the genotype or the coelomic fluid fraction from which they were identified.

In the last experiment, we studied the role of the Trf-mediated cellular and humoral fractions in the phagocytosis of *E. coli*. Although it was demonstrated that SpTrf proteins augment phagocytosis [18], the function of the membranal SpTrf proteins in this process was not addressed. We found that although the coelomocytes alone (without CF) are sufficient for phagocytosis, the absence of CF (humoral fraction) reduces phagocytosis efficiency. We also used the anti-SpTrf antibody mix to block the cell surface PlTrf proteins to determine whether this affected phagocytosis efficiency. Our results demonstrate that the inhibition of cell-surface PlTrf activity results in significant inhibition of phagocytosis. Following both results, we hypothesis that the membrane-bound cellular PlTrfs interact with bacteria-bound humoral PlTrf as part of the phagocytosis process.

Taken together, this study characterizes a diverse PlTrf gene/protein family in the Mediterranean Sea urchin *P. lividus*. It demonstrates the collaboration of distinct cellular and humoral PlTrf-mediated effector arms in mounting specific *P. lividus* immune responses, opsonization of bacteria, and phagocytosis.

## 4. Materials and Methods

### 4.1. Alignment and Phylogenetic Analyses of Trf Sequences

The search for *PlTrf* sequences was performed in four TSA databases: HACU01, GCZS01, GEDS01, GFRN01 and in the Octopus database. Available online on request: http://octopus.obs-vlfr.fr/blast/oursin/blast_oursin.php (accessed on 15 March 2021). *S. purpuratus* and *H. erythrogramma Trf* sequences were compared to the databases using the local BLAST algorithm. Twenty-five sequences from each of the three species were aligned using ClustalW multiple alignments with default parameters and further manual refinement using MEGA X [33]. Phylogenetic analyses of PlTrf sequences were performed using Maximum Parsimony, Neighbor-joining, and Maximum Likelihood analyses with bootstrapping of 1000 repetitions. The best algorithm combination was inferred by MEGAX as Maximum Likelihood with Hasegawa–Kishino–Yano model [32]. Initial trees for the heuristic search were obtained by applying the Neighbor-Joining method to a matrix of pairwise distances estimated using the Maximum Composite Likelihood (MCL) approach. A discrete Gamma distribution was used to model evolutionary rate differences among sites (3 categories (+*G*, parameter = 2.7099)). The tree was visualized as unrooted with the bootstrap value displayed in ITOL program [41]. Twelve translated *PlTrf* sequences were chosen based on their unique aa sequence to generate alignments and to represent all available element patterns.

### 4.2. Sea Urchins

Sea urchins were obtained from the Israeli National Center for Mariculture in Eilat. Urchins were kept in artificial seawater (Red Sea Fish Pharm Ltd., Herzliya, Israel) in a 165-L aquarium in the Molecular Ecology Laboratory at the Ariel University, of which 1/10 volume was replaced every week. The water salinity was kept around 40 ppt and the temperature between 20 and 22 °C. The animals were fed once a week with either fresh or frozen *Ulva lactuca* algae. Unhealthy-looking or dying sea urchins are isolated in a different aquarium to avoid contamination of healthy individuals. Experimental animals were isolated in individual floating plastic cages in a separate compartment in the aquarium.

### 4.3. Bacteria Cultures

*Escherichia coli* (25,922 strain) and mCherry-expressing *E. coli* were kindly provided by Dr. Shiri Navon-Venezia and Ms. Helena Tuchinsky, *Vibrio penaeicida* (51,842 strain) were obtained from ATCC bioresource center (Manassas, VA, USA), commercial lipopolysaccharide (LPS) from *E. coli* was purchased from Sigma-Aldrich (Israel). *E. coli* were cultured in liquid Luria Bertani (LB) medium overnight at 37 °C. *V. penaeicida* bacteria were cultured in Marine Broth media (MB) from Sigma-Aldrich (Israel) overnight at 26 °C. The concentration of bacteria was either calculated based on OD values measured in a Nanodrop NP80 spectrophotometer from Implen (Westlake Village, CA, USA) or manually counted with a hemocytometer. All bacteria were heat-killed for 5 min at 95 °C, washed, and resuspended in aCF (10 mM CaCl_2_, 14 mM KCl, 50 mM MgCl_2_, 398 mM NaCl, 1.7 mM Na_2_HCO_3_, 25 mM Na_2_SO_4_ [12]) and used immediately in experiments.

### 4.4. Preparation of Coelomocytes and Cell-Free CF Samples

CF (~300 μL) was collected from the coelomic cavity of the sea urchin with a 23 G needle and a 1 mL syringe filled with an equal volume of ice-cold calcium-magnesium free seawater (CMFSW-EH; 460 mM NaCl, 10.73 mM KCl, 7.06 mM Na_2_SO_4_, 2.38 mM NaHCO_3_, 70 mM EDTA, 20 mM HEPES; pH 7.4; Terwilliger et al. 2006) and kept on ice until processing. The CF was centrifuged at 500× *g* at 4 °C for 5 min. The supernatant was carefully collected without disturbing the cell pellet and transferred to a separate tube. The cells were carefully resuspended and washed in CMFSW-EH. The supernatant was centrifuged again at 2000× *g* at 4 °C for 5 min to obtain the cell-free CF. The cell-free CF and the washed coelomocytes were used separately for WB and phagocytosis assays.

### 4.5. Protein Extraction

Whole CF, cell-free CF, and washed coelomocytes were lysed using RIPA buffer (Thermo Fisher Scientific, Waltham, MA, USA). The lysates were incubated for 30 min at room temperature with protease inhibitor cocktail III (Calbiochem) with constant vortexing. Alternatively, cells were sonicated for 5 min. Cell debris was pelleted by centrifugation for 10 min at 10,000× *g* at 4 °C. The supernatant was stored in aliquots at −80 °C until use. To equalize the protein quantities before loading on the protein gels, protein concentrations were measured using the Bradford assay and with an NP80 spectrophotometer (Implen, Westlake Village, CA, USA).

### 4.6. His60 Ni-Column Binding Assay

The His60 Ni-column binding assay was performed according to [8]. The whole CF was mixed with 2 mL of lysis buffer (50 mM NaH_2_PO_4_, 300 mM NaCl, 10 mM imidazole, adjusted pH to 8.0 with NaOH, supplemented with 2 mg lysozyme). The mixture was sonicated for 5 min and incubated at room temperature for 10 min. The lysate was centrifuged for 30 min at 10,000× *g* at 4 °C, and the supernatant was collected. His60 Ni Superflow Resin (2 mL; Takara Bio, Japan) was loaded into a gravity column, followed by 10 min incubation with 4 mL lysis buffer. The lysate supernatant was added to the column and incubated with resin for 1 h at 4 °C with gentle shaking after which the flow-through was discarded. Unbound proteins were collected with 4 volumes of lysis buffer. Ni-bound proteins were eluted with 5 volumes of elution buffer (50 mM NaH_2_PO_4_, 300 mM NaCl, 500 mM imidazole, water, adjusted pH to 8.0 with NaOH). The flow-through and eluted proteins were used for subsequent analysis by WB and MS.

### 4.7. Western Blots

Lysate samples with equal quantities of proteins were mixed with 4× Laemmli sample buffer (Bio-Rad), heated at 90 °C for 5 min, and 10–30 μg of protein was loaded per lane on 10% or 12% SDS-PAGE gels. Gel electrophoresis was performed in a fresh running buffer (25 mM Tris, 192 mM glycine, 0.1% SDS, pH 8.3). Gels were subjected to semi-dry transfer with Trans-Blot Turbo™ System (Bio-Rad) onto PVDF membranes (Bio-Rad) with the preprogrammed Bio-Rad 1.5 mm gel transfer program. The membranes were incubated for 1 h in 5% blocking solution (BSA-TBST; 5% bovine serum albumin in 20 mM Tris, 150 mM NaCl, 0.1% Tween20). The blocking solution was replaced with 2.5% BSA-TBST solution containing 1:10,000 dilution of primary Rabbit anti-SpTrf antibodies mixture and gently agitated overnight at 4 °C. The membranes were washed 5 times for 5 min each with TBST and then soaked for 1 h with gentle shaking in TBST containing 1:20,000 dilution of HRP labeled Goat anti-Rabbit-Ig antibody (Abcam, Cambridge, UK). Images were obtained by Quantity One or with ChemiDoc imaging systems (Bio-Rad, Haifa, Israel) and processed with ImageJ2 (FIJI) [42] or the Image Lab 6.1 (Bio-Rad, Hercules, CA, USA) software.

### 4.8. Mass Spectrometry

The eluted and washed protein fractions from the nickel affinity column were separated on an SDS-PAGE gel in a duplicate loading series. The gel was cut according to the duplicated lanes. One gel half was used to identify the band sizes by WB. The other gel half was used to cut gel slices that corresponded to high molecular Trf-positive bands that were identified in the WB. An empty gel slice from the gel corner was used as a background control. Gel slices were processed for MS as follows. The bands were subjected to in-gel slice tryptic digestion, followed by a desalting step. The resulting peptides were analyzed using Waters HSS-T3 column on nanoflow liquid chromatography (nanoAcquity) coupled to high resolution, high mass accuracy mass spectrometry (Q Exactive Plus). The samples were analyzed by discovery mode. The data were processed using Proteome Discoverer version 2.4.1.15, searched against custom databases, to which a list of common lab contaminants was added. The search was carried out with the Byonic search algorithms using the fixed modification of cysteine carbamidomethylation, and the variable modification of methionine oxidation, asparagine- or glutamine-deamidation, and protein N-terminal acetylation. One combined database was generated from the custom-tailored subjected databases of Euechinoidea_proteins_collective (53221 protein entries of all identified to the date proteins for Euechinoidea subclass uploaded from NCBI), P_lividus_translated_EMBOSS (raw translated by all 6 frames peptides untreated while removing all X and * indicate stop codons; 468 entries), Trf_P_lividus_cleaned (53 entries of cleaned manually confirmed sequences).

### 4.9. Bacterial Binding Assay

Assays for PlTrf protein binding to *E. coli* employed 10^9^ live bacteria incubated with either live *P. lividus* coelomocytes or cell-free CF for 30 min at room temperature with gentle shaking. After the incubation, coelomocytes were washed 3 times for 5 min each with CMFSW-EH. Bacteria that were incubated with cell-free CF were pelleted at 10,000× *g* for 5 min and washed 2 times for 5 min each. Both coelomocytes and bacteria were lysed and used for Trf protein profiling by Western blot as described above.

### 4.10. Immune Challenge

100 μL of aCF (10 mM CaCl_2_, 14 mM KCl, 50 mM MgCl_2_, 398 mM NaCl, 1.7 mM NaHCO_3_, 25 mM Na_2_SO_4_, and 10 mM HEPES, pH 7.4) [16] containing 10^6^ heat-killed (95 °C for 5 min) *E. coli* or *V. penaeicida* or 1 μg of LPS per mL of sea urchin CF were injected into 3 or 5 sea urchins in each of the experimental groups. The CF volumes was calculated according to (weight of animal (g) × 0.22 = mL whole coelomic fluid (wCF)) [43]. Equivalent volumes of aCF were injected into control animals (*n* = 3 or 5). Before injections, CF from all animals was used to measure the base level of Trf expression.

### 4.11. FCM and FACS

The CF from challenged and unchallenged animals were collected on days 0 and days 1, 2, 14, 30, 43 post-challenge as described above and evaluated by flow cytometry (FCM). All steps in preparation for FCM were performed on ice. Cells were counted and adjusted to 10^6^ cells/mL. Coelomocytes were washed as described above and pelleted by centrifugation at 500× *g* for 5 min at 4 °C. The pellet was gently resuspended in 100 μL of a staining medium (3.3× PBS with 20 mM HEPES, 2% fetal calf serum, 0.09% sodium azide, pH 7.4) plus a mixture of a 1:300 dilution each of three anti-SpTrf polyclonal antibodies provided by L. Courtney Smith. Control samples were resuspended in a staining medium without the primary antibodies. Samples were incubated for 30 min on ice in the dark, followed by cell pelleting and washing in a 500 μL staining medium. The washed cells were resuspended in 30 μL of staining medium with the 1:250 dilution of Goat anti-Rabbit-Ig secondary antibody conjugated to Alexa Fluor 488 (Thermo Fisher Pierce) and incubated on ice for 30 min in the dark. Cells were pelleted and resuspended in 500 μL staining medium containing 1 μg/mL DAPI for the exclusion of dead cells and subjected to the FCM evaluation or by FACS. The evaluation was performed on the Calibur flow cytometer (Becton Dickinson, Franklin Lakes, NJ, USA) and analyzed by FlowJo 10.6 software (FlowJo LLC, Ashland, OR, USA). Sorting was carried out using MA900 cell sorter (Sony, Tokyo, Japan). Sorted live Trf-positive cells were imaged with Nikon Eclipse fluorescent microscope and with ZEISS LSM900 confocal microscope (Thornwood, NY, USA).

### 4.12. Phagocytosis Assay

Cells and cell-free CF from three sea urchins were used in the phagocytosis assay. The cells were washed two times with cold staining medium and pelleted for 3 min at 500 *g* at 4 °C. Washed cells from each animal were divided into three aliquots and kept on ice until use. Either FITC-stained or mCherry expressing *E. coli* were used for the phagocytosis assays. Bacteria were diluted to a concentration of 10^7^ per ml and divided into three groups. Groups A and C were incubated for 40 min with sea urchin CF at 16 °C in the dark. Group B was incubated in the absence of cells. Group C bacteria were subsequently incubated with antiSpTrf polyclonal antibodies mixture at 1:100 dilution in 100 μL for 30 min in the dark. All bacteria samples (A, B, C) were mixed with the corresponding coelomocyte aliquots of each of three sea urchins and incubated in the dark for 70 min. After the incubation, cells were pelleted for 3 min at 500× *g* and loaded to the hemocytometer for the visual evaluation of phagocytosis using fluorescent microscope Nikon Eclipse with standard fluorescent lamp and relevant filters (model IX81, with Nikon filters). Phagocytosis events were counted blindly (without knowing the treatment type), using the 16 middle squares (*n* = 3) of the hemocytometer for A, B and C phagocytosis treatments. Differences in the percentages of cells with phagocyted bacteria in different treatments were assessed by Anova two-factor and one-factor analyses.

## Figures and Tables

**Figure 1 ijms-22-06639-f001:**
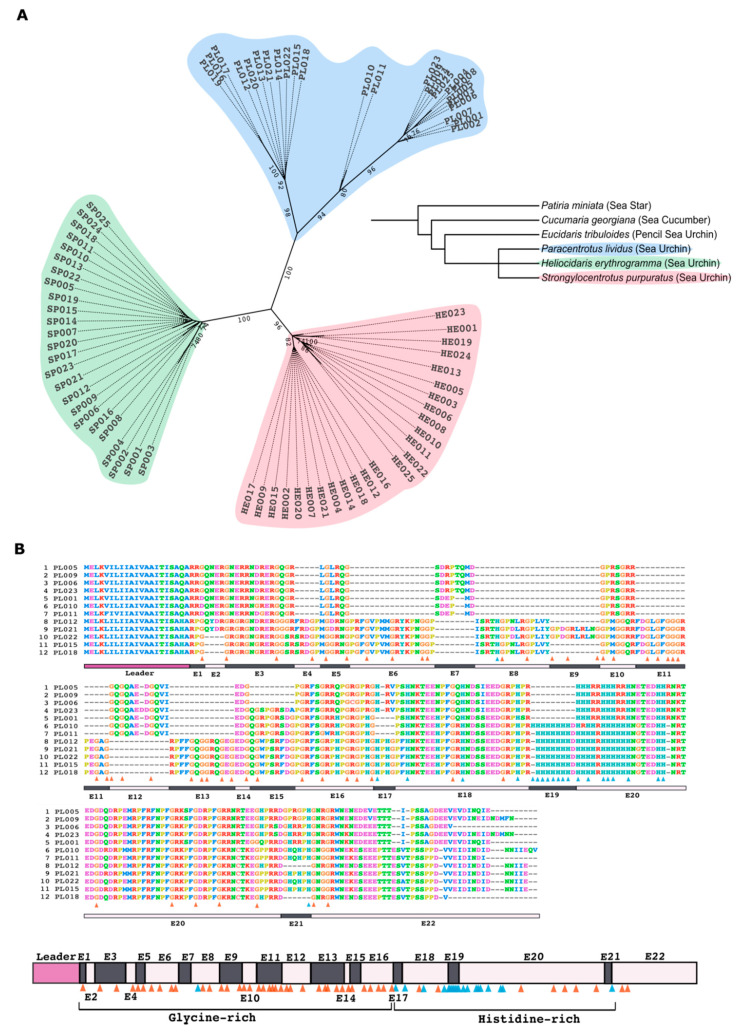
The *PlTrf* phylogeny and the structure of the translated protein sequences. (**A**) An unrooted phylogenetic tree shows separate phylogeny for *PlTrf, HeTrf* and *SpTrf* CDS. Only the ML-based tree (Hasegawa–Kishino–Yano model [32], highest log likelihood (−1640.98) (+*G*, parameter = 2.7099)) is shown. The *Trf* sequences clustered according to the three echinoid species (*P. lividus*, *S. purpuratus* and *H. erythrogramma*). In *P. lividus*, sequences clustered into three clades according to the element patterns. The whole genome-based phylogenetic tree (phyloT, NCBI taxonomy) shows evolutionary relationships among echinoderms species. The cidaroid pencil sea urchin, *Eucidaris tribuloides*, the sea star, *Paritia miniata*, and the sea cucumber, *Cucumaria georgiana*, do not contain *Trf*-like sequences in their genomes. (**B**) Alignment of 12 full-length translated PlTrf aa sequences shows typical Trf structural features. The sequences were aligned in MEGAX Muscle [33] with manual correction based on element size of three or more aa surrounded by gaps. The 12 sequences were chosen from all publicly available *PlTrf* sequences to cover all available sequence element combinations. Sequence alignment is shown. Bars underneath the alignment represent the leader (purple) and the 22 sequence elements (in dark and light shades of gray and light pink). The PlTrf protein structure scheme summarizing the alignment is shown at the bottom. Orange triangles indicate glycine residues. Blue triangles indicate histidine residues. Sequence elements are marked as E1–E22.

**Figure 2 ijms-22-06639-f002:**
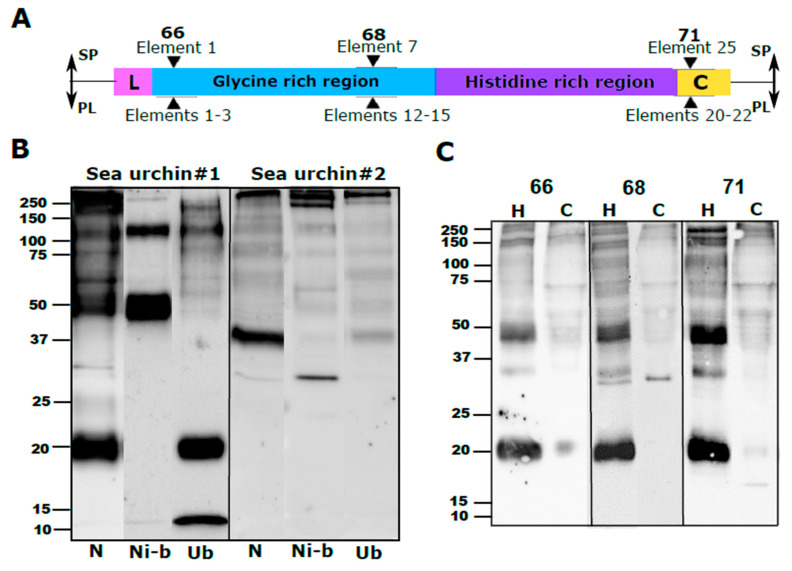
Anti-SpTrf antibodies bind to PlTrf proteins (**A**). Schematic structure of a Trf protein with the locations to which the three SpTrf antibodies bind. The locations and element numbers where the 66, 68, and 71 anti-SpTrf antibodies bind are marked above with black triangles for *S. purpuratus* (SP) and their equivalent elements in *P. lividus* (PL) below. (**B**) WB of native, nickel-bound and unbound proteins shows different subsets of PlTrf proteins. Ni-column PlTrf protein enrichment was performed according to [8], based on binding between the histidine residues and Ni. N, native protein extract from the CF. Ni-b, nickel bound proteins. Ub, unbound proteins that flow through the Ni-column. C. WB of cell-free CF and coelomocytes resulted in similar PlTrf band patterns for each of the three anti-SpTRF antibodies. The WB was made based on a whole protein extract of either CF (H) or coelomocytes (**C**) of the same sea urchin. Each blot was incubated with either 66, 68 or 71 anti-SpTrf antibodies. Differences are observed between the cellular and humoral fractions of CF.

**Figure 3 ijms-22-06639-f003:**
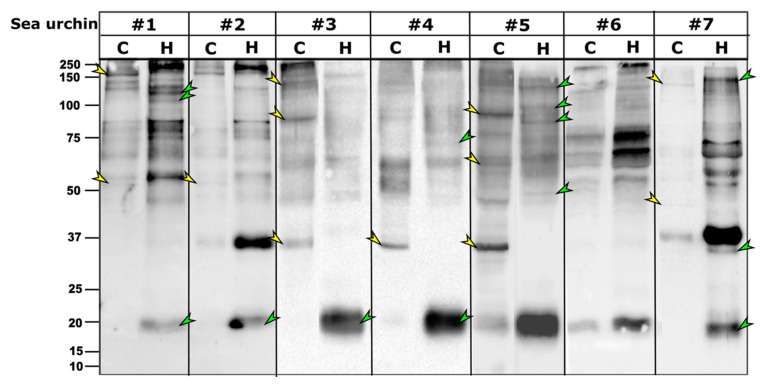
Trf protein repertoires of seven *P. lividus* individuals varies between coelomocytes and cell-free CF. Multiple differences were observed between the cellular and humoral fractions in each of the tested sea urchins, including unique bands in cellular fraction (yellow arrowheads) and unique bands in the cell-free CF fractions (green arrowheads). C. coelomocytes fraction, H. cell-free CF fraction.

**Figure 4 ijms-22-06639-f004:**
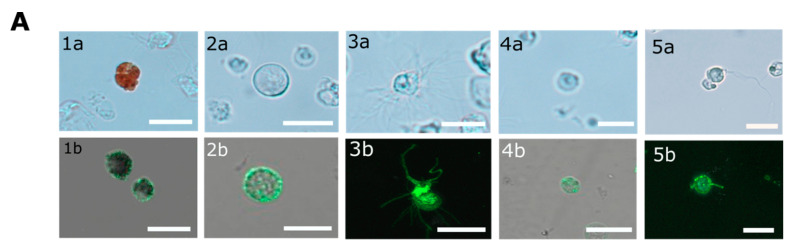
*P. lividus* Trf-positive coelomocytes. (**A**) Cell surface Trf expression in live *P. lividus* coelomocyte of five morphological types. Trf proteins are detected by rabbit anti-SpTrf antibodies and Alexa Fluor 488 secondary goat anti-rabbit-Ig antibody. Trf-positive coelomocytes include: 1. red spherule cells, 2. petaloid or small phagocytes, 3. filopodial phagocytes, 4. colorless spherule cells, and 5. vibratile cells. a, bright field; b, confocal microscope image. Cells in the bright field pictures are not the same cells as in confocal imaging. Scale bars = 10 μm. (**B**) Trf-positive red spherule cells sorting. The APC-positive cells are the red spherule cells. In the α (anti)-SpTrf antibody-treated sample (left), the APC/FITC double-positive cells (upper right corner) are the Trf-expressing red spherule cells. The FITC-positive/APC-negative cells (upper left corner) are all of other SpTrf-positive cells. In the 2nd Ab only control (right) coelomocytes were incubated with the Alexa Fluor 488 secondary anti-rabbit antibody only. Almost no FITC-positive cells were observed in the control.

**Figure 5 ijms-22-06639-f005:**
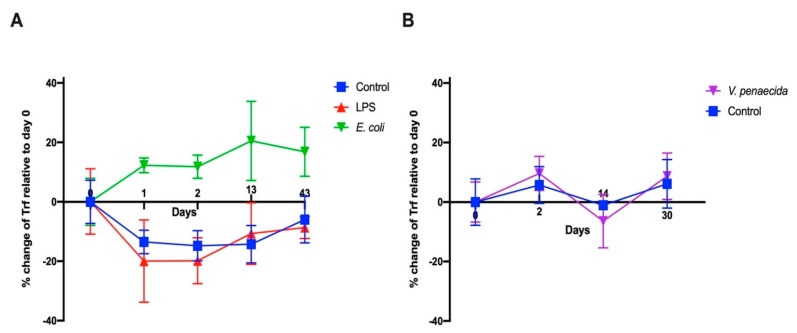
Percent change of Trf-positive cells in response to challenge with different immune elicitors. Due to the high variability in the basal levels of the Trf-positive coelomocytes within the sea urchin population, only the changes in the percentages relative to day 0 are presented. (**A**) The percent change of Trf-positive coelomocytes in response to injections of heat-killed *E. coli*, LPS and aCF (control). Sampling and measurements were carried out on days 0, 1, 2, 13 and 43 post-injection. A significant percent increase of Trf-positive coelomocytes was observed following the challenge with heat-killed *E. coli* over the course of the experiment. (**B**) The percent change of Trf-positive coelomocytes in response to injections of heat-killed *V. penaeicida* and aCF (control). Experiment and control groups did not show significant differences.

**Figure 6 ijms-22-06639-f006:**
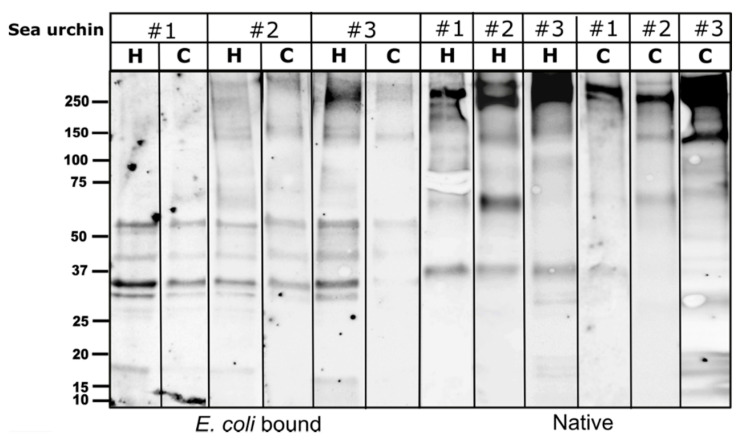
Specific WB band pattern of E. coli bound PlTrf proteins. A similar Trf protein band pattern was observed in both coelomocytes (C) and cell-free CF (CF) from three sea urchins that were incubated with live *E. coli*. The band pattern of the untreated coelomocytes and cell-free CF protein extract counterparts is shown on the right side of the figure. “Native” stands for untreated CF protein extract.

**Figure 7 ijms-22-06639-f007:**
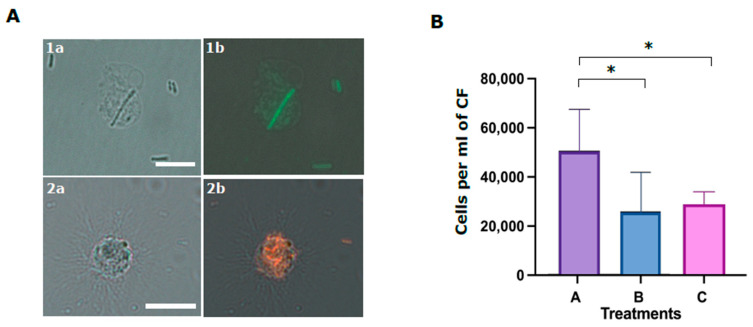
Phagocytosis of *E. coli* bacteria by *P. lividus* phagocytes is inhibited by anti-SpTrf antibodies. (**A**) *P. lividus* sea urchin phagocytes with internalized *E. coli* bacteria. (**1a**,**1b**) A petaloid phagocyte with internalized FITC-stained *E. coli*. (**2a**,**2b**) A filopodial phagocyte with multiple mCherry-expressing *E. coli* in the cytoplasm. Scale bars = 10 μm. (**B**) Inhibition of phagocytosis by SpTrf antibody. Numbers on *Y*-axis represent *E. coli*-containing phagocytes concentration in response to the different treatments. Treatment A. coelomocytes were incubated with bacteria that were pre-incubated with CF. Treatment B. coelomocytes were incubated with untreated bacteria. Treatment C. coelomocytes were pre-incubated with anti-SpTrf antibodies followed by incubation with bacteria that were pre-incubated with CF. Vertical bars indicate standard errors. Differences in *E. coli*-containing phagocyte concentrations in different treatments were significant by Anova two-factor analysis. Anova single factor analysis indicated that treatment A resulted in significantly higher *E. coli*-containing phagocytes concentration compared with treatments B and C. * indicates a significant difference (*p* < 0.05).

**Table 1 ijms-22-06639-t001:** Mass spectrometry PlTrf hits.

	Accession	Name	Coverage [%]	Unique Peptides **	Length aa	MW [kD]	calc. pI
	GEDS01012456.1	Trf [P_lividus]	27	14	301	33.7	7.64
	GCZS01069164.1	Trf [P_lividus]	16	4	243	27.9	8.34
Elution *	GCZS01069162.1	Trf [P_lividus]	4	2	241	27.6	8.46
	HACU01465648.1	Trf [P_lividus]	6	2	244	27.7	6.87
	GFRN01311978.1	Trf [P_lividus]	2	1	239	27.3	8.09
	GEDS01012454.1	Trf [P_lividus]	2	1	307	34.3	8.82
	GEDS01012454.1	Trf [P_lividus]	18	8	307	34.3	8.82
Native *	GFRN01311979.1	Trf [P_lividus]	13	7	239	27.3	8.09
	HACU01465651.1	Trf [P_lividus]	6	2	239	27.2	7.52
	GCZS01069162.1	Trf [P_lividus]	4	2	241	27.6	8.46

* Western blot bands corresponding to the gel slices used for MS analyses are presented in Appendix A. ** Unique peptides are the number of peptides that map uniquely to the protein.

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
