# Peer review of "The Diverse Transformer (Trf) Protein Family in the Sea Urchin Paracentrotus lividus Acts through a Collaboration between Cellular and Humoral Immune Effector Arms"

_ijms, 2021, doi:10.3390/ijms22136639_

Round 1

Reviewer 1 Report

The Manuscript ijms-1256088 entitled “Diverse Transformers (Trf) protein family in the sea urchin Paracentrotus lividus act through collaboration between cellular and humoral immune effector arms” report the characterization of the Trf genes and proteins and the Trf-mediated immune response in Sea urchin species Paracentrotus lividus.

The paper provides data obtained by three different approaches: Western blot analyses for to verify the  cross-reactivity of antibodies with PlTrf proteins; FACS, for identify different coelomocyte sub-populations with detectable membranal PlTrf expression and the role of P. lividus hemolymph in the phagocytosis of E. coli bacteria.

Please find some comments and suggestions below (Minor revisions):

Comments (Minor revision)

Line 96 Add “to” before “an α helical"

Line 321 Add “in” before “the Trf-expressing”

Line 367 “microfilament” not “microfilamental”

Line 380 “phagocyted” not “phagocytosed”

In figures 2C, 3 and 6 we can observe some bubbles that may be an effect of air bubbles between gel and membrane during Western blot transfer step. Do you have any more photos without these bubbles?

Figure 4 Why in this figure some images (1a, 2a, 4a) are presented as merge (bright field/fluorescence) and others only in fluorescence (3a and 5a)?

Author Response

Response to reviewer #1 comments (in red)

Reviewer 1

Line 96 Add “to” before “an α helical"

Fixed 

Line 321 Add “in” before “the Trf-expressing”

Fixed

Line 367 “microfilament” not “microfilamental”

Fixed

Line 380 “phagocyted” not “phagocytosed”

Fixed

In figures 2C, 3 and 6 we can observe some bubbles that may be an effect of air bubbles between gel and membrane during Western blot transfer step. Do you have any more photos without these bubbles?

We agree with Reviewer #1, some pictures contain the visible results of bubbles during the transfer step. We changed where we were able to – in figure 2C for antibody 68. Figure 3 seems to be impaired minorly by the bubbles, so we didn't change it. For figure 6, we, unfortunately, don't have substitutes.

Figure 4 Why in this figure some images (1a, 2a, 4a) are presented as merge (bright field/fluorescence) and others only in fluorescence (3a and 5a)?

Images 3b and 5b were unfortunately only taken in the fluorescent light to construct 3D images (not shown in the article). We chose them for the figure as they are better than other images represent the type of the coelomocyte.

Reviewer 2 Report

This is very welcome paper to the literature on comparative immunobiology of invertebrates. The central issue addressed in the manuscript is a description of diversity of new members from the Transformers protein family – polymorphic and taxa-specific pattern recognition receptors of sea urchins. The manuscript is clearly written, terminology meets accepted standards and references are almost exhaustive. The data were produced in a rigorous and methodologically sound manner. The manuscript has noticeable citation potential and I recommend it for publication after minor revision. My short comments below should be construed as offered in the spirit of trying to improve this paper.

Based on the authors' description, they used calcium-free solutions at almost all steps of their experiments. In particular, the coelomocytes were collected in the isotonic solution (CMFSW) with 70 mM EDTA – extremely high and possibly excessive concentration of the chelating agent. In the phagocytosis assay again, they used Ca-free «cold staining medium». If these methodical descriptions are correct, does this mean that coelomocytic phagocytosis in Paracentrotus lividus is calcium independent? If so, this is very unusual point for marine invertebrates and should be noted in the discussion. Perhaps this finding is even more significant than the description of the functions and diversity of new members from the Transformers protein family.

The authors position sea urchins as “long-living invertebrates”. I suggest to include in the text the longevity estimates (with references) for P. lividus or close related species. This data will definitely increase the interest of ecoimmunologists in their work.

Author Response

Response to reviewer #2 comments

Reviewer 2

Based on the authors' description, they used calcium-free solutions at almost all steps of their experiments. In particular, the coelomocytes were collected in the isotonic solution (CMFSW) with 70 mM EDTA – extremely high and possibly excessive concentration of the chelating agent. In the phagocytosis assay again, they used Ca-free «cold staining medium». If these methodical descriptions are correct, does this mean that coelomocytic phagocytosis in Paracentrotus lividusis calcium independent? If so, this is very unusual point for marine invertebrates and should be noted in the discussion. Perhaps this finding is even more significant than the description of the functions and diversity of new members from the Transformers protein family.

We agree that Calcium is an important factor in phagocytosis. In our phagocytosis assays although we used calcium-magnesium-free seawater for the initial extraction step to avoid clotting, the cell staining medium, which was added later, contained calcium. The recipe is (CMFSW-EH; 460 mM NaCl, 10.73 mM KCl, 7.06 mM Na2SO4, 2.38 mM NaHCO3, 70 mM EDTA, 20 mM HEPES; pH 7.4; Terwilliger et al., 2006)

The authors position sea urchins as “long-living invertebrates”. I suggest to include in the text the longevity estimates (with references) for P. lividus or close related species. This data will definitely increase the interest of ecoimmunologists in their work.

We added “estimated life-span of ~15 years on average” reference - Tomšić, Sanja, et al. "Growth, size class frequency and reproduction of purple sea urchin, Paracentrotus lividus (Lamarck, 1816) in Bistrina Bay (Adriatic Sea, Croatia)." Acta adriatica 51.1 (2010): 67.

Other changes:

We revised the name of the article and instead of “Diverse Transformers (Trf) protein family in the sea urchin Paracentrotus lividus acts through collaboration between cellular and humoral immune effector arms” it is now “The diverse Transformer (Trf) protein family in the sea urchin Paracentrotus lividus acts through a collaboration between cellular and humoral immune effector arms”

We additionally revised the language of the article thanks to the help from Prof. Courtney Smith, a specialist in the field and a native English speaker. We didn't change any results, methods or discussion.